# Dietary Patterns and Gut Microbiota Changes in Inflammatory Bowel Disease: Current Insights and Future Challenges

**DOI:** 10.3390/nu14194003

**Published:** 2022-09-27

**Authors:** Jing Yan, Lei Wang, Yu Gu, Huiqin Hou, Tianyu Liu, Yiyun Ding, Hailong Cao

**Affiliations:** 1Tianjin Key Laboratory of Digestive Diseases, Department of Gastroenterology and Hepatology, Tianjin Institute of Digestive Diseases, Tianjin Medical University General Hospital, Tianjin 300052, China; 2Department of Nutrition, the Second Affiliated Hospital, Air Force Medical University, Xi’an 710038, China; 3Department of Gastroenterology and Hepatology, the Affiliated Hospital of Chengde Medical College, Chengde 067000, China

**Keywords:** inflammatory bowel disease (IBD), food ingredients, gut microbiota, Westernized diet, dietary strategies

## Abstract

Inflammatory bowel disease (IBD) is a result of a complex interplay between genes, host immune response, gut microbiota, and environmental factors. As one of the crucial environmental factors, diet plays a pivotal role in the modulation of gut microbiota community and the development of IBD. In this review, we present an overview of dietary patterns involved in the pathogenesis and management of IBD, and analyze the associated gut microbial alterations. A Westernized diet rich in protein, fats and refined carbohydrates tends to cause dysbiosis and promote IBD progression. Some dietary patterns have been found effective in obtaining IBD clinical remission, including Crohn’s Disease Exclusion Diet (CDED), Mediterranean diet (MD), Anti-Inflammatory Diet (AID), the low-“Fermentable Oligo-, Di-, Mono-saccharides and Polyols” (FODMAP) diet, Specific Carbohydrate Diet (SCD), and plant-based diet, etc. Overall, many researchers have reported the role of diet in regulating gut microbiota and the IBD disease course. However, more prospective studies are required to achieve consistent and solid conclusions in the future. This review provides some recommendations for studies exploring novel and potential dietary strategies that prevent IBD.

## 1. Introduction

Inflammatory bowel disease (IBD) is a multifactorial intestinal disorder characterized by non-infectious chronic and relapsing inflammation of the gastrointestinal tract, and primarily includes ulcerative colitis (UC) and Crohn’s disease (CD). IBD leads to reduced quality of life for patients, and imposes health and economic burdens on families and society worldwide. Epidemiological studies have shown that the incidence in North America and Europe is stable, while the incidence in newly industrialized countries that become more westernized is accelerating [1]. Although the exact etiology of IBD is still unclear, IBD is widely endorsed to be associated with genes, gut microbiota, and environmental factors [2]. Particularly, epidemiological studies have shown that the children of individuals immigrating from regions with low IBD prevalence, such as Middle East, South Asia and Sub-Saharan Africa, to developed regions with high incidence of IBD, had a similar risk of IBD when compared with the offspring of non-immigrants, indicating that risk might be evoked by earlier life exposure to environment [3]. Environmental exposures associated with IBD include breastfeeding, childhood antibiotic exposure, smoking, stress, diet, and physical activity [4], but the most ubiquitous and critical environmental factors are the current Westernized diet and associated microbial dysbiosis [5,6]. In China, the substantially increasing incidence of IBD is congruous with the rapid societal transition to a Westernized diet and culture since the first reported case in 1956 [7]. The modern Westernized diet is usually characterized as calorically dense but nutritionally sparse, with high amounts of protein and (unsaturated) fat, but low amounts of vegetables, fruits, legumes and whole grains [8]. Furthermore, the mounting consumption of ultra-processed foods (UPF) is also associated with a higher risk of IBD onset [9,10]. UPF is similar to the Westernized diet from the point of energy and nutrition, being made from refined carbohydrates, fat, sugar, protein isolates and various additives. The Westernized diet and UPF could reduce gut microbiota diversity and lead to dysbiosis, which is a feature of IBD [11,12]. Hence, current studies aim to improve gut microbiota through diet strategies or food components in order to achieve IBD remission. This review focuses on the current dietary components and strategies aggravating and alleviating IBD and related alteration of gut microbiota.

## 2. Inflammatory Bowel Disease and Gut Microbiota

The human gut harbors 100 trillion of microbes, which contain approximately 100 times more genes than the human genome, forming a complex ecological community and providing immune-modulatory functions through the interactions of microbiotic metabolic activities and the host [13,14,15]. The gut microbiota is dominated by Firmicutes, Bacteroidetes, Actinobacteria, Proteobacteria and Verrucommicrobia phyla, with Firmicutes and Bacteroidetes representing 90%, and it varies with age, genetics, environment, diet and host physiological status [16,17]. The gut microbiota contributes to host nutrition, defense and the development of the immune system. The gut microbiota has considerable power of converting protein into shorter peptides, amino acids and their derivatives, as well as synthesizing vitamin K and B group vitamins including thiamine, riboflavin, pyridoxine, nicotinic acid, pantothenic acid, folates, biotin and cobalamin [18,19]. Additionally, colonic bacteria hydrolyze indigestible dietary fibers such as polysaccharide plant cell walls, soluble oligosaccharides and resistant starch to produce short chain fatty acids (SCFAs), mainly including acetate, propionate, and butyrate [20]. SCFAs regulate the gut metabolism, proliferation and differentiation due to their impact on gene regulation [21]. In particular, butyrate is not only the primary energy source of intestinal epithelial cells, modulating immune and inflammatory responses as well as intestinal barrier function by facilitating the differentiation of regulatory T-cells (Tregs) and effector T-cells, but it also represses cancerous cell expansion [22,23]. Butyrate can be produced by several clusters of *Clostridiaceae* (*Frimicutes*), *Lachnospiraceae*, and *Eubacteriaceae*, potentially *Clostridiaceae*, including: cluster IV (e.g., *Faecalibacterium prausnitzii*, *Butyricicoccus pullicaecorum*, *Subdoligranulum variabile*, *Anaerotruncus colihominis*, and *papillibacter cinnamivorans*), XIVa (e.g., *Eubanterium rectale*, *Roseburia* spp., *Anaerostipes* spp., *Clostridium* spp., *Ruminococcus* spp., *Coprococcus* spp., *Butyrivibrio* spp.), XVI, and I [21,24].

Clinical and experimental data suggest that gut dysbiosis is associated with the occurrence and pathogenesis of IBD. The biodiversity of gut microbiota is diminished in IBD patients, with a lower abundance of anaerobic bacteria (Clostridium cluster IV, XIVa, XVII and *Faecalibacterium Prausnitzzi*) and a higher abundance of Proteobacteria such as *Enterobacteriaceae*, *Bilophila* and certain members of Bacteroidetes [25,26]. The proportion of SCFA-producing bacteria in fecal samples is lower in IBD patients but mucolytic bacteria (*Runinococcus gnavus*, *Blautia gnavus* and *Ruminococcus torques*) and pathogenic bacteria such as *Escherichia coli* (*E. coli*) increase [26,27,28]. Moreover, increased numbers of sulfate-reducing bacteria, such as *Desulfovibrio*, are found in the feces of IBD patients, which produce hydrogen sulfate during the dissimilatory sulfate reduction process, which damages intestinal epithelial cells and induces mucosal inflammation [29].

## 3. Westernized Diet, Gut Dysbiosis and IBD

It is apparent that gut microbiota is influenced by diet. The emergence of the current Westernized diet is related to the change of food production and technology, which makes UPF and foods rich in refined grain, oil and sugar more accessible. The Westernized diet often means a lower intake of dietary fiber from grains, vegetables and fruits and a higher consumption of animal fat, fried food, salt and UPF rich in refined sugars, oils and various food additives, which are often consumed in IBD patients [30]. As mentioned above, Westernized diet-associated gut dysbiosis is recognized as the predominant environmental factor in IBD (Table 1). The Westernized diet tends to result in reduced microbial diversity and clear differences in microbial composition [31,32,33]. It also elevates levels of mucin-degrading bacteria, including *Bacteroides thetaiotaomicron* and *Akkermansia muciniphila* [34]. In addition, a combination of a high-fat and high-sugar diet induced increased Proteobacteria and adherent-invasive *E. coli,* while Firmicutes were more abundant in mice fed a conventional diet [32]. Increased *Bacteroides* spp. and *Ruminococcus torques* were found in mice fed a high-fat and high-sugar diet [33]. Thus, the production of SCFAs by intestinal microbiota decreased, paralleled with decreased Treg population in mesenteric lymph nodes [32]. These changes are found to be detrimental to intestinal mucus and increase the risk of IBD (Figure 1).

### 3.1. High Protein

Some previous studies suggested that high protein intake, especially red meat and processed meat, increased the risk of IBD [46,47,48,49]. A case–control study from Jordan found that IBD patients ate higher amount of dietary protein compared with healthy people [50]. Red meat mainly contains protein, fat and hemoglobin. Several animal studies showed an alteration of gut microbiota composition after high-protein diet, which in turn resulted in changes of metabolites such as amino acids, hydrogen sulfide and branched-chain fatty acids, leading to impaired intestinal barrier function and immune response (Table 1). Moreover, a high-protein diet promotes pro-inflammatory macrophage responses to exacerbate colitis [51]. A high-protein diet increased numbers of *E. coli* while decreasing the numbers and activity of propionate- and butyrate-producing bacteria such as *Bifidobacterium*, *Prevotella*, *Ruminococcus bromii*, and *Roseburia/Eubacterium rectale*, as well as decreased the abundance of *Faecalibacterium prausnitzii* in the large intestine, which had been demonstrated to exhibit an anti-inflammatory effect in a mouse colitis model [35,36]. They also found that a high-protein diet in adults rats for 6 weeks accompanied a reduction of IgG concentration in the colonic lumen. A high-protein diet increased the abundance of *Lactobacillus*, the Family *XIII AD3011 group*, and *Desulfovibrio*, which may be associated with skatole and indole production [37]. Statole and indole are typtophan metabolites, and statole was found to be associated with the pathogenesis of IBD via inducing intestinal epithelial death [52]. Furthermore, a high intake of red meat resulted in reduced abundance of *Lachnospiraceae_NK4A136_group*, *Faecalibaculum*, *Blautia* and *Dubosiella*, and increased abundance of *Bacteroides* and *Alistipes* in dextran sulfate sodium (DSS)-induced colitis [38]. Most of the studies found the negative effect of excessive intake of animal protein, especially red meat, while soy-pea protein was considered having a protective effect [53]. However, a study showed that a high-protein diet, no matter whether derived from casein mainly from animal food or whey from milk or soy, could aggravate acute DSS-induced colitis [54]. They also found that a high-casein diet had no significant effect on the synthesis of Mucin-2 but impaired the mucus barrier by decreasing *Bacteroides thetaiotaomicron* and total mucin-degrading bacteria, which encode sialidase, an enzyme in the first step of mucin degradation [54]. However, two studies have shown that low red meat intake had no protective effect on IBD, without data of gut microbiota changes. A cohort study investigated the dietary intake of fiber, red and processed meat on 56,468 individuals for median 22.2 years and found no relationship between a high intake of fiber and/or a low intake of meat and the risk of a late onset of chronic inflammatory diseases [55]. A randomized controlled trial on CD patients in remission showed no significant differences in relapse (62% to 42%) when consuming high amounts of meat (a minimum of 2 servings/week of red or processed meat) and low amounts of meat (not more than 1 serving per month), respectively, for 49 weeks [56]. Perhaps the patients who reduced red and processed meat replaced it with some other food items that has a deleterious effect on CD. Another two large prospective cohort studies have observed an increased risk of UC for increasing red meat intake [49,57]. While no association was observed in CD patients. In addition, there was no relationship between other food sources of animal protein (including processed meat, fish, shellfish, eggs and poultry) and the risk of CD or UC.

### 3.2. Heme

Heme in red meat may also be pertinent to IBD. Heme iron can be captured by many bacteria and in turn has a profound impact on the microbial community structure. Animal studies have shown that dietary heme significantly decreases microbial diversity and has similar microbial alteration as DSS-treated mice, characterized by reduced Firmicutes and increased Proteobacteria, particularly *Enterobacteriaceae* [58]. However, Khalili et al. have not observed an association between dietary heme iron and risk of CD or UC, and they found that a coding variant in FcγRILA gene, rs 1801274, preferentially modified the association between dietary heme iron and risk of UC [57]. This means that dietary heme probably increases the risk of UC among individuals with a genetic predisposition (rs 1801274-G/G). However, the association of red meat with UC risk was not modified by rs 1801274, suggesting the potential effect of high animal protein on UC risk. Further studies are needed to identify the relationship between high -protein and high-animal-protein diets, dietary heme and IBD.

### 3.3. Fatty Acids

The dietary ratio of *n*-6/*n*-3 polyunsaturated fatty acids (PUFAs) intake has increased from about 1 to 2:1 as humans evolved. Nevertheless, in the Westernized diet, the ratio of *n*-6/*n*-3 PUFAs intake in the diet is up to 20:1 [59]. There have been so many studies demonstrating the association of a high-fat diet and high ratio of *n*-6/*n*-3 PUFAs with a high incidence of IBD and intestinal microbial shift. In a 6 month randomized controlled-feeding trial, 217 healthy young adults were provided with three isocaloric and varying amount of fat diets, containing lower fat (fat 20% energy), moderate fat (fat 30% energy) and higher fat (fat 40% energy). The higher-fat diet was associated with decreased abundance of *Faecalibacterium* and increased *Alistipes* and *Bacteroides*. The concentration of SCFAs in feces significantly decreased but plasma pro-inflammatory factors increased. On the contrary, the lower-fat diet was associated with an increased abundance of *Blautia* and *Faecalibacterium* as well as elevated SCFAs [60]. A high-fat diet (60% energy from fat) rich in *n*-6 PUFAs (~30% total fat) also increased the number of Gram-negative *Enterobacteriaceae* [61]. Another high fat diet rich in *n*-6 PUFAs starting from weaning to adult on mice model induced a significant reduction in Firmicutes, *Clostridia*, and *Lachnospiraceae*, and an increase of Bacteroidetes and *Deferribacteraceae* [39]. Additionally, a high-fat-diet-induced free fatty acid reduced the number of small intestinal intraepithelial lymphocytes and lamina propria lymphocytes, termed “intestinal lipotoxicity”, which was independent of the gut microbes [62]. However, Xie M et al. found that a linoleic acid-rich diet had little effect on the severity of colitis in the spontaneous colitis-prone (IL-10-/-) mice, suggesting a limited role of the increased consumption of dietary *n*-6 PUFAs on promoting IBD [63]. In addition to the high ratio of *n*-6/*n*-3 PUFAs, a recent study has shown that trans-fatty acids, which are commonly present in frying foods and UPF, can exacerbate intestinal inflammation [64]. A trans-fatty-acid-enriched diet increased the abundance of Proteobacteria and *Desulfovibrionaceae* while decreasing the relative abundance of Bacteroidetes, *Lachnospiraceae*, *Bacteroidales S24-7* and decreasing the levels of butyric acid and valeric acid, which might contribute to the development of intestinal inflammation [65,66]. It is also involved in extracellular matrix remodeling of the intestinal mucus., However, the authors surprisingly found an increased abundance of *Lactobacillaceae*. Maybe more studies are needed to clarify the effect of *n*-6 PUFAs and trans-fatty acids on IBD.

### 3.4. High Sugar

It seems that a high intake of refined carbohydrates and sugar is associated with risk of IBD, but the findings are still inconsistent. Sugars are commonly used in sweetened foods such as cakes, cookies, ice creams and beverages, which are popular in the Westernized diet and modern society. It has been found that IBD patients had a higher intake of carbohydrates in the form of carbonated beverages [67]. A recent cohort study of 121,490 participants showed that a increased consumption of sugar-sweetened beverages was associated with significantly higher risk of CD, but this association was not significant for UC patients. Additionally, there was no association between the consumption of artificially sweetened beverages or natural juices and risk of IBD [68]. Some other studies also found a positive association between sugars, soft drinks and risk of IBD [69,70,71,72,73]. However, three meta-analysis suggested no association between total carbohydrate intake and risk of IBD [69,70,74]. This might be explained by the inclusion of dietary fiber in the total carbohydrates intake, which belongs to non-digestible carbohydrates. Maybe a high consumption of sugar and soft drinks is associated with UC risk only if one had low vegetable intakes, demonstrating the importance of dietary fiber [75]. It suggests that the type of dietary carbohydrate is more important than the total amount of carbohydrate in determining the risk of IBD, but there was no association between sugar-sweetened beverages and the risk of IBD in a few studies [70,71]. This may be due to a small proportion of sugar-sweetened beverages in daily use. In addition, IBD patients above 40 years have less sugar-sweetened beverages consumption than younger patients and adolescents, which may affect the study results.

Some experimental studies have shown the negative effect of dietary sugars in IBD by gut microbiota alteration (Table 1). A diet rich in simple sugars significantly increased the abundance of pathobionts, such as *E. coli* and *Candida*, and promoted neutrophil infiltration, leading to the disruption of barrier integrity [40]. In another study, a high glucose or fructose (10%) diet did not trigger inflammatory responses, but significantly altered the gut microbiota, increasing the abundance of the mucus-degrading bacteria *Akkermansia muciniphila* and *Bacteroides fragilis*, which paralleled with enriched bacteria-derived mucolytic enzymes, thus destroying the mucus layer [41]. A study evidenced that high-fructose-diet-induced colitis was microbiota-dependent, with increased *Akkermansia muciniphila* and the pathogenic microbe *Citrobacter Rodentium* [42]. Fructose feeding also led to a remarkably increased abundance of Proteobacteria and decreased Bacteroidetes, and significantly reduced SCFAs [76]. Furthermore, a high-fructose diet directly suppressed the growth of *Bifidobacterium pseudolongum* and *Lactobacillus johnsonii*, which promoted the expression of cholylglycine hydrolase, an enzyme that breaks down conjugated bile acids [42]. Increased conjugated bile acids reduced mucus thickness and worsened colitis. However, they found that ahigh-glucose diet did not exacerbate DSS-induced colitis. This might be attributed to the fact that glucose and fructose are metabolized in different ways not only in the liver but also in the central nervous system. Glucose intake elevates blood glucose levels and inhibits appetite, while fructose is metabolized far more rapidly than glucose by bypassing the rate-limiting regulatory step of glycolysis. Meanwhile, a high-fructose diet increases the hunger hormone ghrelin, promoting appetite and food intake, and also induces inflammatory responses [77]. Similarly, a high-sugar diet was found to increase gut permeability, decrease microbial diversity and reduce SCFAs, with elevated Verrucomicrobia while reducing Firmicutes and Tenericutes at the phyla level; this diet also elevated *Verrucomicrobiaceae* and *Porphyromonadaceae* but decreased *Anaeroplasmataceae*, *Prevotellaceae* and *Lachnospiraceae* at the family level [43]. Therefore, refined sugars and fructose should be restricted in diet to prevent IBD.

### 3.5. Food Additives

With the increased consumption of UPF and concomitantly elevated incidence of IBD, some studies have linked food additives with the development of intestinal inflammation and dysbiosis. For instance, ubiquitous sodium chloride, dietary inorganic phosphate, the emulsifiers sodium stearoyl lactylate, polysorbate 80 (P80) and carboxymethylcellulose (CMC), the coating and thickening agents maltodextrin (MDX), carrageenan and gum, the artificial sweeteners neotame, splenda and sucralose, the colorant titanium dioxide, and the antiseptic agent potassium sorbate were all searched in animal models [44,78,79,80,81,82,83]. For instance, MDX enhances the adhesiveness of the invasive *E. coli* strain LF82, neotame was found to reduce α-diversity in mice, and titanium dioxide impairs the abundance of *Bifidobacterium* and *Lactobacillus*. It has been found that maternal P80 intake even led to gut dysbiosis in offspring and increased their susceptibility to colitis in adulthood [84]. Although the dosage of food additives in experiments may be much higher in daily use, the latent danger should not be ignored, and safer and natural food additives should be discovered and used.

## 4. Dietary Strategies Alleviating IBD

### 4.1. Crohn’s Disease Exclusion Diet (CDED)

Many previous studies have shown that exclusive enteral nutrition (EEN) can induce remission and mucosal healing and is recommended as the first-line therapy in children and adolescents with CD by current guidelines [85]. However, EEN for 6–12 weeks is difficult to perform, which has led to decreased compliance and poorer results, especially in adult patients. Some researchers tried to use Crohn’s Disease Exclusion Diet (CDED) and partial enteral nutrition (PEN) to treat patients who had difficulty in continuing EEN (Table 2). CDED regimen consists of two phases over 12 weeks with incremental categories of food allowed. The first phase lasts for 6 weeks, aiming to induce remission and is more restrictive than the second phase. The food containing dietary components that could hypothetically induce dysbiosis, increase intestinal permeability or degrade the mucus layer are excluded or limited, including condiments, sauces, animal fats, gluten, dairy products, processed meats, packaged products, and products containing certain monosaccharides and emulsifiers. Besides this, fruits and vegetables are restrained to reduce fiber exposure. During phase II, access to food is broadened. A fixed portion of whole grain, fruits and vegetables are allowed to increase the flexibility of the diet and improve adherence [86].

Sigall-Boneh et al. evaluated the efficacy of CDED or CDED (accounting for 50% of dietary calories) combined with PEN in early mild-to-moderate CD in children and young adults. Twenty-seven patients (67.5%) obtained remission in the CDED plus PEN group and six of seven obtained remission in CDED group. [86] They found decreased C-reactive protein in 70% of patients who achieved clinical remission. They also performed a single-center trial of 21 patients (11 adults and ten children) with loss of response to biologics, who received the CDED plus PEN by a polymeric formula for 12 weeks, and 13 (61.9%) achieved clinical remission, demonstrating that CDED plus PEN may be used as a salvage regimen for patients failing biological therapy [87]. A retrospective analysis was performed to assess the efficacy of CDED in comparison with EEN for remission induction. A total of 42 children (68.9%) achieved remission in a total of 61 patients; 27 of 41 (65.9%) received EEN and 15 of 20 (75%) received CDED plus PEN (with prior 1–2 weeks of EEN) [89]. Furthermore, they found that patients with the CDED plus PEN regimen gained more weight compared to patients with EEN alone. In a 12-week prospective trial in 74 children with mild-to-moderate CD, the combination of CDED and PEN was shown to be more effective and better tolerated than EEN [88]. One group received CDED plus 50% of calories from formula for 6 weeks followed by CDED with 25% partial enteral nutrition from weeks 7 to 12, while the other group received EEN for 6 weeks followed by a free diet with 25% partial enteral nutrition from weeks 7 to 12. They found 28 (75.6%) of 37 and 14 (45.1%) of 31 children were in corticosteroid-free remission respectively, with a sustained reduction in serum level of C-reactive protein, fecal level of calprotectin and fecal Proteobacteria. CDED combined with PEN was also demonstrated to be effective on both induction and maintenance in children with mild-to-moderate CD by a case series [112]. In a multicenter randomized trial, 73 children with mild-to-moderate CD were given exclusive enteral nutrition or the CD exclusion diet (CDED) for 3 weeks. A total of 82% of patients in the CDED group and 85% of patients in the EEN group had a rapid response or remission. However, the odds of remission reduced at week 6 due to poor compliance [90]. In another 12 weeks of study, the effectiveness of CDED was evaluated in adult patients [91]. A total of 32 patients with CDED plus 50% PEN obtained 76.7% and 82.1% clinical remission at 6 and 12 weeks respectively. These data demonstrate that CDED is an effective therapy for inducing clinical remission in both children and adult CD patients. In a recent open-label, pilot randomized trial in adults with mild-to-moderate CD, 19 patients received CDED plus partial enteral nutrition and 21 patients received CDED alone, with 13 (68%) and 12 (57%) patients achieving clinical remission respectively at week 6 [92]. A total of 20 (80%) of 25 patients in remission at week 6 sustained remission at week 24, and 14 (35%, 8 patients in the CDED plus partial enteral nutrition and 6 in the CDED group alone) of 40 patients were in endoscopic remission at week 24. These data further demonstrate that CDED with or without partial enteral nutrition was effective for the induction and maintenance of remission in patients with mild-to-moderate CD. However, only one study examined the gut microbiota of patients receiving CDED or CDED plus partial enteral nutrition. Although CDED is easier to implement than EEN, it still needs adherence to rigid dietary protocols, limiting long-term compliance. Moreover, dairy products and red meats are not allowed in CDED, which may increase the risk of inadequate protein and iron intake, and aggravate the existing anemia in some CD patients. In contrast, CDED plus PEN may ensure adequate protein and micronutrients intake. Further studies should not only pay attention to clinical and endoscopic remission, but also the adequacy of energy and nutrients intake.

### 4.2. Mediterranean Diet (MD)

The Mediterranean diet (MD), a typical dietary pattern amid the areas of the Mediterranean, is defined by relative high intake of fresh vegetables, fruits, unprocessed cereals, olive oil, nuts and legumes, a moderate-to-high consumption of fish, moderate intake of dairy products (mostly as cheese and yogurt), as well as a lower consumption of sweets, meat and meat products [113,114,115,116]. A prospective cohort study performed in 83,147 subjects between 45 and 79 years of age confirmed 164 cases of CD and 395 cases of UC with an average follow-up of 17 years (1997–2017). This study found that a greater adherence to MD was associated with a significantly lower risk of CD but not UC, and sex, age, educational level, body mass index and smoking could not modify these associations [113]. Moreover, adherence to MD is associated with a lower level of fecal calprotectin in IBD patients, both in children and adults [114,115]. Nutritional education may help promote greater adherence to MD thus improving quality of life and modulating disease activity [116]. Existing data indicate that the MD exhibits favorable microbiota profiles, which may influence the occurrence of IBD (Table 3). For instance, subjects who adhered to the MD with a higher consumption of fruits, vegetables and legumes were associated with an increased level of fecal SCFAs and some fiber-degrading bacteria, and a greater presence of Bacteroidetes was related to lower animal protein intake [117,118]. Individuals with a higher adherence to the MD had higher total bacteria and SCFAs, lower amounts of *E. coli* and a higher *Bifidobacteria*:*E. coli* ratio [119]. MD intervention also resulted in increased levels of the fiber-degrading *Faecalibacterium prausnitzii* and of genes involved in microbial carbohydrate degradation, which is connected with butyrate metabolism [120]. *Faecalibacterium prausnitzii* possesses anti-inflammatory properties and is reduced in active IBD patients [121]. In contrast, a lower adherence to the MD was linked to a higher ratio of Firmicutes–Bacteroidetes and a higher urinary level of trimethylamine oxide, a potential risk factor of colorectal cancer. Illescas O et al. reported an increase in *Akkermansia* and a reduction in *Fusobacterium* in MD, even below the levels observed in healthy subjects without a defined diet [122]. *Akkermansia* is a marker of a healthy gut, and *Fusobacterium* is a known pathogenic bacterium associated with cancer and IBD. Collectively, these studies support diet diversity and remind us that the sufficient inclusion of a variety of plant-based food is more important than the exclusion of animal-based food.

Abundant amounts of vegetables and fruits is a property of MD, which are not only rich in dietary fibers and micronutrients, but also the source of biophenols. Pomegranate, curcumin, bilberry anthocyanin, apple, mango, naringenin, ginger, and green tea showed potential therapeutic effects for IBD in some interventional studies [127,128]. In addition, olive oil is a special component in MD compared with other diet patterns and accounts for the majority of the antioxidant and anti-inflammatory activity of MD [128]. A study showed that MD enriched with 40 g/d high quality-extra virgin olive oil increased adiponectin, IL-10 concentration and lactic acid bacteria, while decreasing markers of inflammation and oxidative stress such as myeloperoxidase and 8-hydroxy-2-deoxyguanosine after being given to 18 overweight/obese and 18 normal weight subjects for three months [129]. Studies in vivo and vitro showed that olive oil supplementation and olive-oil-derived biophenols significantly reduced the secretion of pro-inflammatory cytokines such as IL-1β, IL-6 and tumor necrosis factor-α (TNF-α), repaired the membrane oxidative damage and thus maintained the epithelial integrity [128,130,131]. Several studies have demonstrated that olive-oil-derived biophenols consumption increases gut microbiota diversity and *Bacteroidetes* and/or reduces the *Firmicutes**/*Bacteroidetes** ratio. It also increases some beneficial bacteria such as *Bifidobacteria* and *Lactobacillus*, and promotes the production of SCFAs [123].

The biophenols of olive oil include both simple structures such as phenolic acid and phenolic alcohols, and complex structures, such as flavonoids, secoiridoids, and lignans [128]. Oleuropein, the major phenolic secoiridoid of olive tree leaves, also exhibited anti-inflammatory activity in colonic biopsies from UC patients [132]. Oleuropein-treated colonic biopsies from UC patients showed significantly decreased CD3, CD4 and CD20 cells and reduced level of cyclooxygenase-2 (COX-2) and IL-17 compared with those treated with lipopolysaccharide from *Escherichia coli* alone [132]. Oleuropein can be considered as an inhibitor of several gastrointestinal pathogens and also a kind of prebiotics to be used as a carbon source of *Lactobacillus* and *Bifidobacterium* strains [123,133]. When oleuropein was loaded in nanostructured lipid carriers, it showed to be a more effective way to decrease colonic inflammation and enhance reactive oxygen species activity, providing a potential strategy for the treatment of UC [134].

Adherence to the MD suggests an increased consumption of long-chain *n*-3 PUFAs derived from fish, seafood and nuts, and also relatively higher tissue *n*-3 PUFA levels [135]. Many studies have explored the relationship between fish consumption, dietary *n*-3 PUFA and IBD, but the findings were inconsistent. Huang X et al. found that the habitual consumption of fish oil was associated with a 12% lower risk of IBD and 15% lower risk of UC [136]. Mozaffari H et al. analyzed five prospective and seven case–control studies including 282,610 subjects with 2002 IBD patients (1061 CD and 937 UC) and found that a higher consumption of fish was associated with a lower incidence of CD, especially in Asian countries, and higher dietary long chain omega-3 PUFAs was associated with lower risk of UC. However, there was no association between fish consumption and UC, as well as between α-linolenic acid and IBD [137]. The comprehensive results of this meta-analysis suggest that the influence of different nations, dietary patterns and sub-type of fish and *n*-3 PUFAs on the risk of IBD should be taken into consideration. Nevertheless, many studies evaluated the efficacy of *n*-3 PUFAs on the resolution of IBD as a safer and well-tolerated alternative treatment. Indeed, *n*-3 PUFAs have been demonstrated to decrease the *Firmicutes/Bacteroidetes* ratio but increase SCFA-producing bacterial genera, such as *Bifidobacterium*, *Roseburia* and *Lactobacillus* [124,125]. However, *n*-3 fatty acids supplementation is not recommended to support the maintenance of remission in patients with IBD in ESPEN guidelines due to its ineffectiveness in previous systematic reviews [85]. In recent studies, dietary *n*-3 PUFA was found to attenuate colitis by decreasing colon inflammatory markers such as IL-6, COX-2 and TNF-α in rodent models, but has no effect on tight junction proteins [138]. Eicosapentaenoic acid (EPA), a major component of *n*-3 PUFA, was used to demonstrate its anti-inflammatory ability in IBD. A placebo-controlled trial with UC patients given either EPA–FFA (500 mg, twice daily) or placebo for 6 months showed that EPA decreased fecal levels of calprotectin without serious adverse events, which may provide an agent to induce and maintain symptom-free remission in patients with UC [139]. However, in a study from Schwärzler J, both *n*-3 and *n*-6 PUFAs were found to instigate epithelial chemokine production by activating endoplasmic reticulum sensor inositol-requiring enzyme 1α [140].

In addition to exploring the effects of olive oil and *n*-3 fatty acids on IBD, there are studies evaluating the effect of adherence to MD on disease activity and inflammatory markers. El Amrousy D et al. compared clinical score and inflammatory markers in MD and a normal diet group for children and adolescents with mild-to-moderate active IBD, and found significant decreases in CRP, calprotectin, TNF-α, IL-17, IL-12, IL-13 and clinical scores in the MD group [93]. Another study compared the Specific Carbohydrate Diet (SCD) and MD as treatment for mild-to-moderate CD patients, who received prepared meals and snacks according to their assigned diet in the first 6 weeks and then followed their own diet in the second 6 weeks, and found the SCD was not superior to the MD to achieve symptomatic remission (SCD, 46.5%; MD, 43.5%; P = 0.77), nor to reduce calprotectin or CRP [107]. The MD may be a potential dietary pattern for patients who are poorly compliant to special dietary patterns that have more restrictions in terms of food, and further studies are required to test its efficacy for IBD patients in Asia.

### 4.3. Low-FODMAP Diet

The low-“fermentable oligo-, di-, mono-saccharides and polyols” (FODMAP) diet, defined as the restriction of foods high in fermentable oligosaccharides (fructans, galacto-oligosaccharides), disaccharides (lactose), monosacchrides (fructose), and polyols (sorbitol, mannitol, xylitol), is often used in irritable bowel syndrome patients because it may reduce symptoms of bloating, cramping and diarrhea [141]. The degree of carbohydrate digestion and absorption is influenced by interindividual variation, the dose consumed and some diseases. Apart from some short-chain fermentable carbohydrates being absorbed such as fructose and lactose, the remaining unabsorbed carbohydrates lead to increased small intestinal water (fructose, lactose and polyols) and gas production after being fermented in the colon, thus inducing functional gastrointestinal symptoms [142]. It has been estimated that at least one-third of quiescent IBD patients experience functional gastrointestinal problems, which has promoted many studies probing the effect of a low-FODMAP diet on IBD [143]. Some studies considered that a low-FODMAP diet could reduce gastrointestinal symptoms in patients with quiescent IBD, especially abdominal bloating, followed by diarrhea, abdominal pain, fatigue and nausea, but had no effect on constipation [144,145]. In addition, a low-FODMAP diet may ameliorate disease activity, decrease pro-inflammatory markers such as fecal calprotectin and C-reactive protein, and improve the quality-of-life of IBD patients in remission or with mild disease activity [95,146,147] (Table 2). Another study also showed patients who received a low-FODMAP diet had a greater amelioration in irritable bowel syndrome severity and quality of life but had no significant improvement on markers of inflammation [96]. However, a low-FODMAP diet may often greatly restrict the consumption of many fruits and vegetables, resulting in a relatively lower amount of fiber and reduced growth of beneficial bacteria and SCFA production. Adherence to a low-FODMAP diet reduces luminal *Bifidobacteria* according to the literature [148]. In fact, findings about the effect of a low-FODMAP diet on gut microbiota were inconsistent. Cox et al. showed significant reduced abundance of *Bifidobacterium adolescentis*, *Bifidobacterium longum*, and *Faecalibacterium prausnitzii*, but with no differences in microbiome diversity in low-FODMAP diet group [96]. Halmos et al. showed increased abundance of *Ruminococcus torques* and decreased abundance of *Clostridium cluster XIVa* and *Akkermansia muciniphila* with no difference of SCFAs and total bacterial abundance during a low-FODMAP diet intervention [94]. In parallel, long-term adherence to the low-FODMAP diet might cause a risk of malnutrition in IBD patients due to a lower fiber consumption paired with a reduced energy intake [149], because the average energy intake amount is 1696 kcal/day in the low-FODMAP diet group compared to 1918 kcal/day in the control group [96]. Therefore, the existing studies failed to examine the actual effectiveness of the low-FODMAP diet due to lacking adequate evidence in terms of its quality and quantity, and it should not be recommended for IBD patients with functional gastrointestinal symptoms [150]. Further studies with objective outcomes such as immune activation markers, gut microbiota and mucus integrity are required to support the generalization of this approach for clinical practice in IBD patients.

### 4.4. Anti-Inflammatory Diet

Some studies have suggested that adherence to a pro-inflammatory diet was associated with a greater possibility of IBD, but the results were not ubiquitous [151,152,153,154]. In order to evaluate the inflammatory nature of a person’s diet, some methods calculating the inflammatory index are proposed. The Dietary Inflammatory Index (DII) scores are determined by 45 food parameters such as various types of fat, micronutrients, alcohol, anthocyanidins, flavones and some special foods like garlic, ginger and pepper [155], while Empirical Dietary Inflammatory Index (EDII) scores are calculated according to 18 food groups, including processed meat, red meat, refined grains, beverages and tomatoes, which are positively associated with inflammation, and beer, tea, coffee, and dark yellow and leafy green vegetables, which are inversely associated with inflammation [156]. DII and EDII scores, both positive (>0) and negative (<0), reflect pro- and anti-inflammatory diets, respectively. DII was found to be positively associated with disease activity in CD patients, and was significantly different between patients in remission and with mildly and moderately active disease [152]. However, the association was not significant in UC patients. Vagianos K et al. investigated the impact of a change in diet on changes in fecal calprotectin and IBD symptoms using the DII and EDII [153]. The results showed that an increase in the EDII score was associated with higher fecal calprotectin and more severe symptoms. The association between the DII score and calprotectin or IBD symptoms was not observed, suggesting that the DII might not adequately determine the relationship between inflammatory nature of a diet and IBD. Interestingly, a cross-sectional study showed neither the DII nor EDII score was associated with disease activity [154]; this could be because not all food parameters in DII can be assessed with the food frequency questionnaire or found in the food composition database, and not all the foods are included in the EDII. Furthermore, different sample sizes may lead to different results. The inflammatory potential of diet also influences gut microbiota. Adherence to a pro-inflammatory diet was found to increase the abundance of *Ruminococcus torques*, *Eubacterium nodatum*, *Acidaminococcus intestini* and *Clostridium leptum*, and this tendency became more evident when adjusting for age and body mass index, while adherence to an anti-inflammatory diet lead to enriched *Akkermansia muciniphila* [45]. However, overall, diversity did not significantly differ.

In order to solve nutrition adequacy, malabsorption and gastrointestinal symptoms, the IBD-Anti-Inflammatory Diet (IBD-AID) was developed to offer to patients who were refractory to pharmacological therapy, or could not comply with long-term exclusion enteral nutrition [157]. IBD-AID mainly contains five components.

(1)Restrict specific carbohydrates such as refined or processed complex carbohydrates and lactose.(2)Increase the consumption of prebiotics, probiotics and food rich in components that help restore the balance of intestinal flora.(3)Increase foods intake rich in omega-3 PUFAs while reducing total fat and saturated fatty acids intake.(4)Evaluate patient’s dietary pattern and monitor potential nutrient deficiencies. Modify food texture to improve nutrient absorption (e.g., homogenized, cooked, ground) [157].

The studies on IBD-AID are severely limited. In a retrospective study, 11 adult patients with IBD who received AID for 4 or more weeks all had symptom reduction and were able to stop at least one of their previous drugs [157]. In another study, 25 children with active CD were assigned to either EEN or PEN (75% of dietary needs) and one meal from AID for 6 weeks. Clinical remission rates were 69.2% and 75%, and mucosal healing rates were 45.5% and 27.3% in the EEN and PEN groups respectively, with the same endoscopic remission rate of 45.5%, demonstrating the validity of PEN plus AID in inducing clinical and endoscopic remission in children with active CD [158]. Although AID accounted for only a small part of the whole energy intake, this provided evidence for the application of AID in IBD treatment. Keshteli AH et al. designed a 6 month, open-label, randomized, placebo-controlled trail in adult UC patients. Patients in the AID group showed a higher subclinical response and increased levels of fecal *Bifidobacteriaceae*, *Lachnospiraceae*, and *Ruminococcaceae* compared with that of “Canada’s Food Guide” group. In addition, patients in the AID group had significant changes in the metabolome, including decreased fecal acetone and xanthine levels and increased fecal taurine and urinary carnosine and p-hydroxybenzoic acid levels, which contributed to the increased intake of flavonoids and seafood [97]. However, more prospective studies are needed to demonstrate its efficacy. As the AID requires the restriction of processed complex carbohydrates, it may increase the risk of inadequate calorie intake and weight loss, and not all the patients prefer whole-grain foods. Both patients with CD and patients with UC were more likely to consume more refined grains in practice, which increases the difficulty of AID implementation. Furthermore, restriction of dairy products has the potential of leading to calcium and vitamin D deficiency and contributes to the inadequate intake of protein. Campmans-Kuijpers MJE et al. designed the Groningen anti-inflammatory diet (GrAID) based on evidence of effects of food (groups) and dietary patterns on the onset and course of IBD, which contained more food (groups) than other dietary patterns such as MD, IBD-AID, and the low-FODMAP diet [159]. In GrAID, wheat, fruit, vegetables, legumes, lean meat, fish, eggs, plain dairy (milk, yoghurt, kefir and hard cheeses), coffee, tea, and honey are allowed, red meat, sugar and other dairy products are limited, while alcohol, sweetened beverages, and canned and processed foods are avoided, which can provide IBD patients with evidence of selecting beneficial foods and avoiding detrimental food in the course of medical treatment [159]. The effect of GrAID and the gut microbiota alteration after AID need to be tested in the future.

### 4.5. Specific Carbohydrate Diet (SCD)

The specific carbohydrate diet (SCD), initially used for celiac disease in the mid-20th century, and then used in IBS and IBD, is hypothesized to decrease intestinal inflammation by changing pro-inflammatory gut microbes that are perpetuated by poorly absorbed carbohydrates, mainly disaccharide and polysaccharide carbohydrates. Thus, grains are restricted including wheat, barley, corn and rice, and nut flours such as almond flour and coconut flour are used to make bread and other baked goods. Patients eat monosaccharide carbohydrate such as glucose and fructose from honey and most fruits. Most milk products are also limited except for fully fermented yogurt. The foods allowed include most fresh fruits and vegetables, meat, yogurt, nuts and hard cheeses. A prospective study showed that the nutrient intake of pediatric IBD patients receiving SCD was adequate under close monitoring [160]. Six of eight individuals had gained weight, one had weight loss, and one had no change in weight. Energy intake was adequate, and vitamins B2, B3, B5, B6, B7, B12, C, A, and E met or exceeded the recommended daily allowance (RDA) for most patients, while the intake of vitamin D was below the RDA in all the participants, and calcium intake was below the RDA in 75% of the patients. Moreover, homemade SCD tends to provide more adequate energy than chef-prepared SCD [161].

The majority of the studies evaluating the efficacy of SCD are retrospective or case series studies conducted in the pediatric population and mainly in CD patients, and found decreased C-reactive protein (CRP), hematocrit, albumin or erythrocyte sedimentation rates (ESR), as well as an improvement in disease activity [98,99,100,101,103,104] (Table 2). For many patients, after a period of strict SCD, restrictions are eased off once the symptoms and laboratory markers are improved. Some SCD “illegal” food are gradually added to the diet, such as rice, oats, potatoes, and quinoa, which is called the modified SCD (mSCD) [104]. Wahbeh GT et al. compared the mucosal healing effect of the SCD and mSCD in seven pediatric CD patients for 26 months on average, and found complete macroscopic mucosal healing was not seen in any patient and fecal calprotectin was mildly elevated, in spite of normalized CRP, albumin and hematocrit [104]. Weight and height gain were analyzed in pediatric CD patients on SCD and the majority of children had both weight and height gain, providing another subjective evidence for the efficacy of SCD [159]. Another special study was an anonymous online survey containing 417 patients on the SCD with IBD [102]. A total of 47% of patients had CD, 43% had ulcerative colitis, and 10% had indeterminate colitis, and 56% took medication in addition to the SCD. Additionally, 13% of patients reported time to achieve remission of less than 2 weeks, 17% reported 2 weeks to a month, 36% reported 1–3 months, and 34% reported greater than 3 months, suggesting an overall benefit for patients on the SCD with significant heterogeneity. However, this survey could not evaluate disease activity. It was quite possible that patients still had ongoing inflammation although they felt clinically improved. Subsequently, Suskind DL et al. found there was no clear pattern of dysbiosis in IBD patients before the SCD, which was corrected after the SCD intervention [105]. They also compared the efficacy of SCD, mSCD (SCD with oats and rice), and a whole-food (WF) diet eliminating wheat, corn, sugar, milk and food additives [106]. PCDIA, CRP and ESR all decreased in three groups, but calprotectin decreased only in the mSCD and WF groups, which suggested that polysaccharide carbohydrates were not as harmful as expected. Although the nature of the microbiome composition changes was largely patient-specific, the abundance of some bacterial populations in each patient changed 10-fold or more over the course of the treatment. The abundance of a *Blautia* species, a *Lachnospiraceae* species, *Faecalibacterium prausnitzii*, *Roseburia hominis*, *Roseburia intestinalis*, *Anaerobutyricum hallii*, and *Eubacterium eligens* increased in at least four of the five patients, while the abundance of *E. coli* and a strain of *Faecalibacterium prausnitzii* decreased in more than three patients. However, the property of microbiota alteration needs further study due to small sample sizes used. As the SCD requires the restriction of processed complex carbohydrates, it may increase the risk of inadequate calorie intake and weight loss, and not all the patients prefer whole-grain foods. Both patients with CD and patients with UC were more likely to consume more refined grains in practice, which increases the difficulty of SCD implementation. Furthermore, the restriction of dairy products has the potential to lead to calcium and vitamin D deficiency and contributes to an inadequate intake of protein. A recent study showed that SCD was not superior to the MD [107] and patients felt greater ease in following the MD, which means that the dietary patterns should not only be effective, but also easy to adhere to.

### 4.6. Plant-Based Diets

Plant-based diets encourage the consumption of nutrient-dense plant foods including vegetables, fruits, beans, peas, lentils, soybeans, seeds, and nuts, while limiting processed foods, oils, and animal foods (including dairy products and eggs) [162]. Plant-based diets include various types such as lacto-ovo vegetarian, lacto-vegetarian, ovo-vegetarian, vegan, pescatarian and semi-vegetarian [163]. Although a study found vegetarian diet was protective factor for UC but risk for CD, a plant-based diet has shown nutritional benefit, in particular increased fiber, vitamin C and K, beta carotene, magnesium, and potassium intake [164]. Chiba M et al. performed a series of studies evaluating the effect of plant-based diet combined with infliximab or other medications [108,109,110,111] (Table 2). For CD patients with clinical remission, the semi-vegetarian diet showed a 100% remission rate at 1 year and 92% at 2 years [108]. For CD patients, a lacto-ovo-semi-vegetarian diet combined with infliximab decreased disease activity and CRP, and achieved mucosal healing in 46% of patients [109]. For severe UC patients, the lacto-ovo-semi-vegetarian diet combined with medications significantly decreased CRP and ESR at week 6, and achieved 76% in the remission rate and 6% in the colectomy rate [111]. Furthermore, the cumulative relapse was 25% with no additional colectomy at 1 year follow-up. For UC patients in an initial episode and relapse, the lacto-ovo-semi-vegetarian diet combined with medications achieved 14% and 27% in cumulative relapse rate, respectively, at 1 and 5 years follow up for the initial episode cases, and 36% and 53% respectively for relapse cases [110]. It has been shown that anti-TNF therapy shifted the diversity of fecal microbiota in patients with IBD toward that of healthy individuals, but levels of butyrate and substrates involved in butyrate synthesis in fecal samples were significantly reduced in non-responders, which suggested the insufficient consumption of dietary fiber in non-responders [165]. Plant-based diets, low in animal protein and fat and rich in dietary fiber and polyphenols, increase microbial diversity, enhance the abundance of beneficial bacteria such as *Bifidobacterium*, *Lactobacillus*, and *Faecalibacterium prausnitzii* and produce beneficial microbial metabolites such as SCFAs, while inhibiting the proliferation of pathogen-associated bacteria [126], which may help to decrease the rate of nonresponse. However, plant-based diets may result in an inadequate intake of vitamin B12 and other micronutrients depending on the strictness of the diet, seasonal effects (vitamin D status) and the nutritional status of the patients, which may affect the effectiveness of plant-based diets on IBD. In a study, 1254 IBD patients reported clinical data and dietary habits between 2006 and 2015, and 4.1% (*n* = 52) of the patients followed a vegetarian diet (VD) and 4.7% (*n* = 54) a gluten-free diet (GFD). Patients on a VD or GFD had no difference in disease activity, fistula, hospitalization or surgery rates, but had significantly higher levels of post-traumatic stress symptoms. They thought patients on a VD or GFD tended to have lower psychological well-being [166]. Therefore, an appropriate plant-based diet pattern may be an important factor.

### 4.7. Other Dietary Patterns

Other potential dietary strategies include gluten-free diet, intermittent fasting, low-fat, high-fiber diet, Paleolithic diet and Jiangnan diet, but the evidence about these diets for the treatment of IBD are insufficient or lacking.

The gluten-free diet is often used in individuals with wheat allergy, celiac disease, and non-celiac gluten sensitivity [167]. Gluten is a mixture of gliadins and glutenins found in most grains such as wheat, rye, and barley. Some animal studies showed a gluten-containing diet exacerbated mucosal damage by damaging adhesion junctions and desmosomes as well as shortening microvilli and modifying endocytic vesicle route [168]. Gliadin and amylase trypsin inhibitor, the components of gluten, were also found to result in a pro-inflammatory immune response and alteration of fecal microbiota [167,169]. Although almost one-third of the IBD patients suffer from gluten sensitivity, there is no causal relationship between gluten and IBD [170]. A recent prospective cohort study found that dietary gluten intake was not associated with risk of IBD [171]. In addition, there have been no prospective studies evaluating the effect of gluten-free diet on the induction and maintenance of IBD and alteration of gut microbiota.

Intermittent fasting, a dietary approach in which individuals repeatedly, voluntarily and severely restrict food intake for about 16 to 24 h, has emerged to decrease inflammatory status [172]. Recent animal studies showed that intermittent fasting, including time-restricted fasting and intermittent energy restriction, suppressed inflammatory responses and oxidative stress in colon tissue, and promoted the regeneration and repair of damaged intestinal epithelium [173,174]. It also altered gut microbiota such as increasing the abundance of anti-inflammatory microbes and decreasing the enrichments of colitis-related microbes such as *Shigella* and *E. coli* [173]. Fasting or calorie restriction could even induce autophagy in many tissues and organs [175]. However, Negm M et al. recruited 80 patients with IBD to observe the effect of Ramadan intermittent fasting on disease activity, which is part of the religious rituals of Muslims. They found that the Mayo score had significantly risen after intermittent fasting, especially in older patients and those with higher baseline calprotectin levels, but serum CRP and stool calprotectin did not change significantly before vs. after fasting [176]. The therapeutic potential of intermittent fasting on IBD needs more human studies.

The Low Carbohydrate Diet (LCD) and Ketogenic Diet (KD) are very popular in the scientific community and general public in the last decades due to their effectiveness for weight loss. The LCD is usually defined as a lower proportion of total daily calories derived from carbohydrates (<20%) with a relatively high percentage of daily calories from protein (25–30%) and fat (55–65%) [177]. KD means an even lower amount of carbohydrates, i.e., less than 50 g of carbohydrate per day. LCD was found to exacerbate DSS-induced colitis and increase the level of *Escherichia/Shigella*, while KD alleviated colitis and increased the level of *Akkermansia* [178]. KD alters the gut microbiota in a manner different from a high-fat diet and elevates circulating ketone bodies, which inhibit the growth of *Bifidobacterium* and reduce the level of intestinal pro-inflammatory Th17 cells [179]. However, in another study, KD was found to aggravate DSS-induced colitis with increased pathogenic bacteria such as *Proteobacteria*, *Enterobacteriaceae*, *Helicobacter* and *Escherichia–Shigella* and decreased potentially beneficial *Erysipelotrichaceae* [180]. Moreover, the actual role of the KD on human beings with IBD remains unknown, and the health effects of long-term KD also require further studies.

Jiangnan diet, a dietary pattern in the Yangtze River Delta region in China, is characterized by consuming abundant vegetables and moderate fruits, especially dark green leafy vegetables, more high-quality white meat rather than ret meat, a larger portion of soybean products and a smaller portion of dairy products, and sometimes a little millet wine [181]. In addition, people in Jiangnan region tend to take more varieties of dishes with small portions, avoiding excessive energy intake. The Jiangnan diet might be a good choice for those who had difficulties in achieving high compliance to follow MD, but there has been no evidence for the effect of the Jiangnan diet on IBD and gut microbiota yet.

## 5. Conclusions

There is compelling evidence to demonstrate that diet is an essential regulatory factor of the gut microbiota, which in turn is associated with many inflammatory disorders including IBD. The Westernized diet and some dietary components are found to be linked with an increased incidence of IBD and dysbiosis. Hence, gradually increasing studies put emphasis on diet modification due to its lower costs and fewer side effects. Some dietary strategies have been found effective in improving disease activity, even achieving and maintaining clinical remission; some others need further prospective evidence. Notably, an effective dietary pattern that works for some patients may not work for others, and a balanced diet is more recommended because it not only avoids certain nutrients deficiencies, but is also easy to adhere to over time. For patients treated with medication, an appropriate diet can also enhance the effectiveness of medication. Therefore, more studies are needed to identify effective dietary measures fit to different patients. It is also an important direction for future research to further investigate the mechanisms of specific dietary components that promote the development of IBD or obtain remission, in order to provide a theoretical basis for precise intervention.

## Figures and Tables

**Figure 1 nutrients-14-04003-f001:**
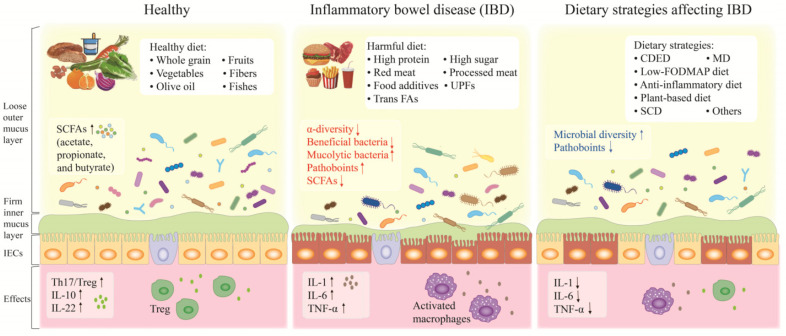
The role of diet in the pathogenesis and remission of inflammatory. Studies have demonstrated that the Westernized diet and certain dietary components alter the gut microbiota, intestinal mucosal layer as well as mucosal immunity, which are associated with initiation of IBD. In contrast, some food composition and dietary patterns protect or improve intestinal microbiota and result in IBD remission. ↑ represents an increase, ↓ represents a decrease. Abbreviations: IECs, intestinal epithelial cells; PUFAs, poly-unsaturated fatty acids; SCFA, short chain fatty acids; UPFs, ultra-processed foods; FAs, fatty acids; CDED, Crohn’s disease exclusion diet; MD, Mediterranean diet; FODMAP, fermentable oligo-, di-, mono-saccharides and polyols; SCD, specific carbohydrate diet.

**Table 1 nutrients-14-04003-t001:** Characteristics of the Westernized diet and pro-inflammatory diet and gut microbiota alteration.

Diet	Characteristics	Changes of Gut Microbiota	References
Westernized diet	High-fat and high-sugar	Proteobacteria ↑ *E. coli* ↑ *Bacteroides* spp. ↑ *Ruminococcus torques* ↑	[32,33]
	High protein	*E. coli* ↑ *Bifidobacterium* ↓ *Prevotella* ↓ *Ruminococcus bromii* ↓ *Roseburia* ↓ *Eubacterium rectale* ↓ *Faecalibacterium prausnitzii* ↓ *Lactobacillus* ↑ *XIII AD3011 group* ↑ *Desulfovibrio* ↑	[35,36,37]
	High red meat	*Bacteroides* ↑ *Alistipes* ↑ *Lachnospiraceae_NK4A136_group* ↓ *Faecalibaculum* ↓ *Blautia* ↓ *Dubosiella* ↓	[38]
	High fat rich in *n*-6 PUFAs	Bacteroidetes ↑ *Deferribacteraceae* ↑ Firmicutes ↓ *Clostridia* ↓ *Lachnospiraceae* ↓	[39]
	High sugar	*E. coli* ↑ *Candida* ↑ *Akkermansia muciniphila* ↑ *Bacteroides fragilis* ↑ *Citrobacter Rodentium* ↑ *Bifidobacterium pseudolongum* ↓ *Lactobacillus johnsonii* ↓ *Verrucomicrobiaceae* ↑ *Porphyromonadaceae* ↑ *Anaeroplasmataceae* ↓ *Prevotellaceae* ↓ *Lachnospiraceae* ↓	[40,41,42,43]
	More food additives	α-diversity ↓ *Bifidobacterium* ↓ *Lactobacillus* ↓ Pathoboints ↑ Mucus-degrading bacteria ↑	[44]
Pro-inflammatory diet	More pro-inflammatory foods	*Ruminococcus torques* ↑ *Eubacterium nodatum* ↑ *Acidaminococcus intestine* ↑ *Clostridium leptum* ↑ *Akkermansia muciniphila* ↑	[45]

↑ represents an increase, ↓ represents a decrease. Abbreviations: PUFAs, polyunsaturated fatty acids; *E. coli*, *Escherichia coli.*

**Table 2 nutrients-14-04003-t002:** Studies on diet strategies alleviating inflammatory bowel disease.

Diet Regimen	First Author	Study Design	Population	Intervention (Duration)	Control Group	Key Findings	Changes of Gut Microbiota
**Crohn’s disease exclusion diet (CDED)**	Sigall-Boneh R (2014) [86]	R	34 children and 13 adults with mild-to-moderate CD	PEN + CDED (12 weeks, *n* = 40) CDED (12 weeks, *n* = 7)	N/A	Clinical remission observed in 24/34 children and 9/13 adults at week 6 and maintained in 27/33 patients at week 12; Significant fall in clinical disease activity and C-reactive protein.	Not analyzed
	Sigall-Boneh R (2017) [87]	R	11 adults and ten children with loss of response to biologics in CD	PEN + CDED (12 weeks, *n* = 21) Paediatric pateints with severe flares: EEN (14 days); PEN + CDED (10 weeks)	N/A	Clinical remission obtained in 13/21 (62%); Decrease in CRP and Harvey Bradshaw Index and increase in albumin.	Not analyzed
	Levine A (2019) [88]	P	74 children with mild to moderate CD, six withdrawed	PEN (50%) + CDED (6 weeks); PEN (25%) + CDED (6 weeks, *n* = 40)	EEN (6 weeks); PEN (25%) + free diet (6 weeks, *n* = 38)	Clinical remission observed in 28/37 received CDED plus PEN and 14/31 received EEN and then PEN; Sustained fall in serum level of C-reactive protein, fecal level of calprotectin.	*Haemophilus* ↓ *Veillonella* ↓ *Bifidobacterium* ↓ *Prevotella* ↓ *Anaerostipes* ↓ *Oscillibacter* ↑ *Roseburia* ↑
	Niseteo T (2022) [89]	R	61 children	EEN (1–2 weeks) CDED + PEN (*n* = 20)	EEN (*n* = 41)	Clinical remission observed in 15/20 received CDED + PEN and 27/41 received EEN; Higher weight gain in CDED + PEN group.	Not analyzed
	Sigall Boneh R (2021) [90]	P	73 children with mild to moderate CD	CDED + PEN (6 weeks, *n* = 39)	EEN (6 weeks, *n* = 34)	Rapid response or remission observed in 32/39 received CDED + PEN and 29/34 received EEN at week 3.	Not analyzed
	Szczubełek M (2021) [91]	P	32 adults	PEN (50%) + CDED (12 weeks)	N/A	Clinical remission observed in 76.7% patients at 6 weeks and 82.1% at 12 weeks. Significant fall in calprotectin level.	Not analyzed
	Yanai H (2022) [92]	P	40 adults	CDED (24 weeks, *n* = 21)	CDED + PEN (24 weeks, *n* = 19)	Clinical remission observed in 12/21 received CDED alone and 13/19 received CDED plus PEN, endoscopic remission observed in 6/21 received CDED alone and 8/19 received CDED plus PEN.	Not analyzed
**Mediterranean diet**	El Amrousy D (2022) [93]	P	100 children and adolescents with mild/moderate active IBD	MD (12 weeks, *n* = 50)	Regular diet (12 weeks, *n* = 50)	Significant decrease in PCDAI, PUCAI and inflmmatory markers (CRP, calprotectin, TNF-α, IL-17, IL-12 and IL-13)	Not analyzed
**Low-FODMAP diet**	Halmos EP (2016) [94]		9 patients with clinically quiescent CD	Low-FODMAP diet (21 days) received low or tipical FODMAP diet with ≥21 day washout	N/A	Symptoms relief in low-FODMAP diet, but no effect on calprotectin.	*Ruminococcus torques* ↑ *Clostridium cluster XIVa* ↓ *Akkermansia muciniphila* ↓ no difference in SCFA and total bacterial abundance
	Bonidi G (2019) [95]	P	55 adults with IBD (38 CD/22 UC)	Low-FODMAP diet (6 weeks, *n* = 26)	Standard diet (6 weeks, *n* = 29)	Disease activity, median calprotectin decreased, and disease-specific quality of life significantly increased in Low-FODMAP diet group but not in the standard diet group.	Not analyzed
	Cox SR (2020) [96]	P	52 patients	Low-FODMAP diet (4 weeks, *n* = 27)	Control diet (4 weeks, *n* = 25)	Adequate relief in gut symptoms received low-FODMAP diet (14/27, 52%) than the control diet (4/25, 16%); Greater reduction in irritable bowel syndrome severity scores and higher health-related quality of life scores received low-FODMAP diet	*Bifidobacterium adolescentis* ↓ *Bifidobacterium longum* ↓ *Faecalibacterium prausnitzi* ↓
**Anti-inflammatory Diet (AID)**	Keshteli AH (2022) [97]	P	53 patients with UC	AID (6 months, *n* = 26)	Canada’s Food Guide (6 months, *n* = 27)	Higher subclinical response (FCP < 150 µg/g at the endpoint) in AID group(69.2 vs. 37.0%)	*Bifidobacteriaceae* ↑ *Lachnospiraceae* ↑ *Ruminococcaceae* ↑
**Specific Carbohydrate Diet (SCD)**	Suskind DL (2014) [98]	R	7 children with CD	SCD (5 to 30 months)	N/A	All patients’ PCDAI decreased to 0 after 3 months; Improvement to normalization of albumin, CRP, and hematocrit.	Not analyzed
	Cohen SA (2014) [99]	P	10 children with active CD (PCDAI ≥ 15)	SCD + prescribed medications (52 weeks)	N/A	Improvement of PCDAI in 9 patients who completed the initial 12-week trial; Continued improvement of PCDAI in 7 patients who maintained 52 weeks and mucocal healing in 2 patients.	Not analyzed
	Khandalavala BN (2015) [100]	Case series	36 patients with CD 9 patients with UC 5 patients with indeterminate colitis	SCD or SCD + medications (mean time 35.4 months) SCD	N/A	Mean effectiveness of 91.3% in controlling acute flare symptoms; Mean effectiveness of 92.1% at maintaining remission.	Not analyzed
	Obih C (2016) [101]	R	20 children with CD 6 children with UC	SCD (3 to 48 months)	N/A	Fall in PCDAI from 32.8 ± 13.2 to 20.8 ± 16.6 by 4 ± 2 wk, and to 8.8 ± 8.5 by 6 months; For in mean PUCAI from 28.3 ± 10.3 to 20.0 ± 17.3 at 4 ± 2 wk, to 18.3 ± 31.7 at 6 months.	Not analyzed
	Suskind DL (2016) [102]	Anonymous online survey	417 patients with IBD (47% CD, 43% UC, 10% indeterminate colitis)	SCD (34.9 ± 16.4 years)	N/A	Clinical remission less than 2 weeks in 13% patients, 2 weeks to a month in 36% patients, 1–3 months in 36% patients, and greater than 3 months in 34% patients.	Not analyzed
	Burgis JC (2016) [103]	R	11 pediatric patients with CD	SCD simple (diet alone, antibiotics or 5-ASA) for 7.7 ± 4.0 months (range 1–12)	SCD with immunomodulators (corticosteroids and/or stable thiopurine dosing)	Improvement in hematocrit, albumin and ESR in both groups; Weight and height gain in the majority of children.	Not analyzed
	Wahbeh GT (2017) [104]	R	7 pediatric patients with CD	Modified SCD (mSCD, 26 months)	N/A	No active systoms before mSCD; Consistent normalization in CRP, albumin and hematocrit; Mild elevation in fecal calprotectin; No endoscopic mucosal healing in any patients.	Not analyzed
	Suskind DL (2018) [105]	P	9 pediatric patients with CD and 3 pediatric patients with UC	SCD (12 weeks)	N/A	Decrease in CRP, PCDAI and PUCAI; No clear pattern of dysbiosis in all patients before the SCD; Correction of dysbiosis in most patients after dietary change.	Not analyzed
	Suskind DL (2020) [106]	P	18 pediatric patients with mild/moderate CD	SCD (12 weeks, *n*-5) Modified SCD (mSCD 12 weeks, *n* = 6)	Whole foods diet (WF 12 weeks, *n* = 5)	Decrease in CRP, PCDIA, ESR in all groups; decrease in calprotectin in mSCD and WF groups; Changes of the microbiota composition showed largely patient specific; Increase in predicted metabolic mode of the organisms.	*Blautia species* ↑ *Lachnospiraceae species* ↑ *Roseburia hominis* ↑ *Roseburia intestinalis* ↑ *Anaerobutyricum hallii* ↑ *Eubacterium eligens* ↑ *E. coli* ↓ *Faecalibacterium prausnitzii* ↓
	Lewis JD (2021) [107]	P	194 patients with mild/moderate CD	SCD (12 weeks, *n* = 97)	MD (12 weeks, *n* = 97)	No difference in symptom remission, calprotectin and CRP. Greater ease to follow the MD	Not analyzed
**Plant-based diet**	Chiba M (2010) [108]	prospective single-group	22 adult CD patients with clinical remission	semi-vegetarian diet (*n* = 16, 2 years)	Omnivorous diet (*n* = 6, 2 years)	100% in remission rate at 1 year and 92% at 2 years in semi-vegetarian diet group.	Not analyzed
	Chiba M (2017) [109]	prospective single-group	46 patients with CD (35 adults and 11 children)	A lacto-ovo-semivegetarian diet combined with infliximab (6 weeks, *n* = 46)	N/A	Decrease in CDAI score and CRP level; Mucosal healing achieved in 46% of cases.	Not analyzed
	Chiba M (2019) [110]	prospective single-group	92 UC (51 initial episodes, 41 relapses)	A lacto-ovo-semivegetarian diet combined with medication	N/A	Cumulative relapse rate rates at 1 and 5 years follow up (Kaplan-Meier analysis) were 14% and 27% respectively for the initial episode of case, and 36% and 53% respectively for relapse cases.	Not analyzed
	Chiba M (2020) [111]	prospective single-group	17 patients with severe UC	A lacto-ovo-semivegetarian diet combined with infliximab (4 years, *n* = 17)	N/A	76% in remission rate and 6% in colectomy rate in the induction phase; Decrease in CRP and ESR at week 6; 25% in cumulative relapse and no colectomy at 1-year follow-up.	Not analyzed

↑ represents an increase, ↓ represents a decrease. Abbreviations: R, retrospective study; P, prospective study; N/A, no applicable; PEN, partial enteral nutrition; PCDAI, Pediatric Crohn’s disease activity index; PUCAI, Pediatric ulcerative colitis activity index; CRP, C reactive protein; ESR, erythrocyte sedimentation rate; MD, Mediterranean diet; AID: anti-inflammatory diet.

**Table 3 nutrients-14-04003-t003:** Characteristics of the Mediterranean diet and plant-based diet and gut microbiota alteration.

Diet	Characteristics	Changes of Gut Microbiota	References
MD	MD adherence	*Bifidobacteria* ↑ *Faecalibacterium prausnitzii* ↑ *Akkermansia* ↑ *E. coli* ↓ *Fusobacterium* ↓	[119,120,122]
	Rich in Olive oil	Bacteroidetes ↑ Firmicutes/Bacteroidetes ratio ↓ *Bifidobacteria* ↓ *Lactobacillus* ↓	[123]
	Rich in *n*-3 PUFAs	*Bifidobacterium* ↑ *Roseburia* ↑ *Lactobacillus* ↑ Firmicutes/Bacteroidetes ratio ↓	[124,125]
Plant-based diet	low in animal protein and fat and rich in dietary fiber and polyphenols	microbial diversity ↑ SCFAs ↑ *Bifidobacterium* ↑ *Lactobacillus* ↑ *Faecalibacterium prausnitzii* ↑	[126]

↑ represents an increase, ↓ represents a decrease. Abbreviations: MD, Mediterranean diet; *E. coli*, *Escherichia coli*; PUFAs, polyunsaturated fatty acids; SCFAs, short chain fatty acids.

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
