# Peer review of "Dietary Patterns and Gut Microbiota Changes in Inflammatory Bowel Disease: Current Insights and Future Challenges"

_nutrients, 2022, doi:10.3390/nu14194003_

Round 1
Reviewer 1 Report
This narrative review aims to summarise current dietary strategies to address IBD and alterations in the gut microbiota. The paper deals with a topical issue. The paper has shortcomings that prevent it from being published in this form.
1. Although it is a narrative review, the method of researching the papers needs to be better defined. It is possible that some papers in the literature were not included and that the papers were chosen randomly.
2. It might be helpful to make the review clearer to use the PRISMA method with an accompanying flow-chart.
3. The tables are difficult to read. The results should be summarised through a better breakdown of the different outcomes and the use of symbols such as arrows for better and easier reading.
4. The discussion should be organized more systematically. It is very easy to read the paper. There is also a lack of references to possible implications in clinical practice and hints for new studies on the subject.
Reviewer 2 Report
This manuscript has good scientific merit and interpretation/discussion and correlation are appropriate. However, some general issues need to be addressed and sentence/and typing/technical errors need to be revised.
My first and foremost suggestion in this review is that it is required to rewrite the abstract, no need to add the reported portion in the abstract of your review and focus to write whatever you covered in this review. The abstract should summarize the complete review.
The following sections also need minor corrections/revisions as per the following comments:
Improve the English language and also check the review for typos and grammar.
Line 29 in the abstract portion this line is looking incomplete please modify it: This review also provides study recommendations of novel and potential dietary strategies that can prevent IBD.
Figure 1 is not clearly visible and doesn’t copy the figure at any place try to make a figure by your own.
Line 85-87: Write the scientific name of bacteria in italics
Line 91-96: Write the scientific name of bacteria in italics: such as Escherichia coli.
Line 259: e Error! Reference source not found.??
Line 493: Error! Reference source not found.??
Line 642: e Error! Reference source not found.??
Line 293 and 295: Improve the sentence language or reframe the sentence and add the reference at the end of the sentence.
Line 126: Write to best of my knowledge that would be more precise to write this sentence.
Line 401-402: Modify the sentence.
Reframe the paragraph 4.5 others: modify or rewrite this section title looks incomplete.
Please re-write some of the words in reference 18: Magnúsdóttir.
Recheck the references section for some more errors.
Round 2
Reviewer 1 Report
The authors improved many parts of the paper following previous suggestions.
It would be good to include a paragraph on low-carb diets and their deleterious effects on gut health, particularly in patients with IBD.
